# Effect of Reactive SPS on the Microstructure and Properties of a Dual-Phase Ni-Al Intermetallic Compound and Ni-Al-TiB_2_ Composite

**DOI:** 10.3390/ma13245668

**Published:** 2020-12-11

**Authors:** Paweł Hyjek, Iwona Sulima, Piotr Malczewski, Krzysztof Bryła, Lucyna Jaworska

**Affiliations:** 1Institute of Technology, Pedagogical University of Krakow, Podchorążych 2 str., 30-084 Krakow, Poland; iwona.sulima@up.krakow.pl (I.S.); piotr.malczewski@up.krakow.pl (P.M.); krzysztof.bryla@up.krakow.pl (K.B.); 2Faculty of Non-Ferrous Metals, AGH University of Science and Technology, 30-059 Krakow, Poland; ljaw@agh.edu.pl

**Keywords:** NiAl alloys, intermetallic matrix composite, FAST/SPS, microstructure, physical, mechanical and tribological properties

## Abstract

As part of the tests, a two-phase NiAl/Ni_3_Al alloy and a composite based on this alloy with 4 vol% addition of TiB_2_ were produced by the reactive FAST/SPS (Field Assisted Sintering Technology/Spark Plasma Sintering) sintering method. The sintering process was carried out at 1273 K for 30 s under an argon atmosphere. The effect of reactive SPS on the density, microstructure, and mechanical and tribological properties of a dual-phase Ni-Al intermetallic compound and Ni-Al-TiB_2_ composite was investigated. Products obtained were characterized by a high degree of sintering (over 99% of the theoretical density). The microstructure of sinters was characterized by a large diversity, mainly in regard to the structure of the dual-phase alloy (matrix). Compression tests showed satisfactory plastic properties of the manufactured materials, especially at high temperature (1073 K). For both materials at room temperature, the compressive strength was over 3 GPa. The stress–strain curves were observed to assume a different course for the matrix material and composite material, including differences in the maximum plastic flow stress depending on the test temperature. The brittle-to-ductile transition temperature was determined to be above 873 K. The research has revealed differences in the physical, mechanical and tribological properties of the produced sinters. However, the differences favourable for the composite were mostly the result of the addition of TiB_2_ ceramic particles uniformly distributed on grain boundaries.

## 1. Introduction

Ni-Al alloys are characterized by high melting point, low density, high thermal conductivity, very good corrosion resistance and resistance to wear, and so the properties are extremely interesting for industrial use [1,2]. This is particularly important in all cases where the mere nature of the operating conditions makes parts produced from metallic or ceramic materials exposed to the risk of frequent failures and damages. Unfortunately, in spite of the fact that these materials exhibit satisfactory mechanical properties at elevated temperatures, they tend to be brittle at room temperature [3,4,5,6]. This factor, which limits their use, can be partially solved by reducing the grain size or producing a two-phase (β + γ’) alloy [7,8,9,10]. One of the promising ways of producing fine-grained intermetallics is powder metallurgy and its combustion synthesis (CS) or self-propagating high-temperature synthesis (SHS) [11,12].

Of particular importance in the production of fine-grained materials is the type of the sintering process selected for practical application, remembering that its main aim is to produce material at the lowest possible temperature and in the shortest possible time, mainly in order to avoid grain growth. One of the recently developed methods that seems to meet the above requirements is spark plasma sintering (SPS). In this method, a pulsating electric current and simultaneously applied pressure are used to create an effective bond between the sintered particles. Due to the Joule effect, the pulsating electric current contributes to the generation of an increased heat flow and removal of the previously absorbed oxides, which act as a barrier to proper consolidation of the powder material [13,14,15,16,17,18].

On the other hand, CS is a technique that uses high exothermic energy to generate high temperature and rapid reaction of combustion. In recent years, CS has been successfully applied in the production of numerous materials, including ceramics, intermetallic compounds, and composites. The high popularity of the CS technique is due to the short production time of these materials (usually in the order of a few seconds) and low energy consumption [15,18,19,20,21,22,23].

Numerous studies have proven that modification of the β-NiAl microstructure with plastic γ’ phase significantly increases the strength properties at both room and elevated temperatures [10,24,25,26]. To provide the best distribution of γ’ phase in the β phase matrix, martensitic transformation of NiAl as a result of hardening and aging of β’-NiAl martensite was applied [27].

The aim of this study was to use, in an unconventional way, an advanced method of sintering to produce a composite material based on intermetallic Ni-Al phases, being aware of the fact that scarce knowledge on this subject is available in the literature. Application of the SPS method to alloys based on the intermetallic phase of NiAl/Ni_3_Al can improve the physical and mechanical properties of the resultant materials. The method has already been used with some success in the manufacture of NiAl [12] and steel matrix composite [28].

In previous studies [29], it has been observed that the addition of TiB_2_ to the two-phase NiAl/Ni_3_Al matrix increases the composite strength and tribological properties. However, in previous studies, other powders (commercial, produced after the exothermic reaction), another sintering method—high-pressure high-temperature (HPHT) and other sintering parameters (T = 1779 K and 7.5 GPa) were used. On the other hand, using the SPS method, the increase in microhardness and compressive strength was observed as a result of the addition of TiB_2_ to the steel matrix [28].

In this study, an attempt was made to evaluate the effect of 4 vol% TiB_2_ addition and CS reaction taking place during SPS in the nickel–aluminium alloy on the microstructure and physical, mechanical, and functional properties of this alloy.

## 2. Research Procedure

The following test materials were used: nickel powder of <150 μm particle size and 99.99% purity, aluminium powder of 100 μm particle size and 99.7% purity, and TiB_2_ powder of 2.5–3.5 μm particle size and 99.9% purity. Figure 1 shows the particle morphologies of individual powders. What attracts attention is the different particle size of nickel (Figure 1a) from what the producer has stated in the sieve analysis. Determined by numerous measurements (optical microscope with image analysis software), the average particle size is 4.5 μm.

Powders of nickel and aluminium in a molar ratio of 1.75:1 were mixed in an air atmosphere with steel grinding media (balls added in a weight ratio of 2:1 relative to the charge) for 20 h. Mixing was performed at 46 RPM (revolutions per minute) in a TURBULA model T2F Shaker Mixer, manufacturer: Willy A. Bachofen AG (Muttenz, Switzerland). The resulting mixture was used as a base material for the manufacture of the sintered matrix and, after adding a suitable amount of the ceramic particles, for the manufacture of a composite with 4 vol% TiB_2_ content. From these mixtures, in a 20 mm diameter graphite die and under an argon protective atmosphere, at a constant anvil pressure of 15 kN, sintered materials were produced using pulsating electric current, the temperature of 1273 K and the test time of 30 s. An HP D5 device from FCT Systeme GmbH (Frankenblick, Germany) was used for this purpose. After proper mechanical treatment of the sample surface, parameters such as density and porosity, Young’s modulus, and Vickers HV1 hardness were measured. The devices used for this purpose included a RADWAG AS 220/C/2 (Radom, Poland) balance with density determination kit, Panametrics EPOCH III ultrasonic flaw detector ((Billerica, MA, USA) and NEXUS 4000 hardness tester (INNOVATEST EUROPE BV, Maastricht, The Netherlands), respectively.

The compression tests were carried out on an INSTRON TT-DM machine (Norwood, MA, USA) with a strain rate of 2.66 × 10^−4^ s^−1^ and at temperatures of 296 K (Room Temperature), 873 K, 973 K and 1073 K. These tests were performed using specimens of 2.5 mm diameter (d) and 3.75 mm height (h), prepared in accordance with the ASTM E9 standard.

The ball-on-disc abrasive wear tests were carried out according to ASTM G99-05 using an ELBIT tribotester (Koszyce Małe, Poland). The test parameters are shown in Table 1. The coefficient of friction, furrow size (scar depth) and the width of the wear track were determined. The depth of the wear scar allowed the determination of the wear rate *V_Z_* in accordance with the following formula:(1)$$V_{Z}=\frac{d_{w}}{t}\;(µm/s)$$
where:*d_w_*—the depth of the wear scar;*t*—test duration.

The wear coefficient *K* was determined according to the formula proposed by Archard:(2)$$K=V/N\cdot l\;({\mathrm{mm}}^{3}/N\cdot m)$$
where:*V*—the volume of material worn;*N*—the load applied;*l*—the sliding distance.

Measurements of the sample/ball mass before and after the test allowed the relative mass loss to be determined from equation:
(3)$$W_{r}=\frac{\Delta m}{m_{o}}\cdot 100\%$$
where:∆*m* = *m_o_* − *m_k_*;*m_o_*—the initial mass of the sample (before the friction test);*m_k_*—mass of the sample after the friction test.

Phases were identified by the X-ray diffraction technique (XRD) using a Dron-3 device (Moscow, USSR). The following parameters were used in the tests: radiation Cu Kα, 38 kV, 30 mA, 20° < 2θ < 100°, 0.02°. Additional evaluation of the sinter quality included examinations of the surface and microstructure of samples and balls after tribological tests, using an Olympus GX-51 optical microscope (Tokyo, Japan) and a JEOL JSM 6610LV scanning electron microscope (Tokyo, Japan) for this purpose.

## 3. Results and Discussion

### 3.1. The SPS Process and Phase Identification

During the FAST/SPS sintering process, when the temperature increases at a rate of 85 K/min, at 848 K, in the Ni/Al powder mixture and in the composite with 4 vol% TiB_2_, the exothermic reaction is initiated (a peak on the temperature curve, sudden change in the values of the plunger force and movement—Figure 2a). Each time, sintering was completed at 1273 K, i.e., after the lapse of 30 s since the plunger movement stabilization (full compaction). As a result of the exothermic reaction occurring between Ni and Al, a two-phase NiAl alloy structure was formed (Figure 3). It was composed of Ni_3_Al (JCPDS 03-065-3245) and NiAl (JCPDS 04-004-7105) phases, as confirmed by X-ray examinations (Figure 2b). In the composite, these studies additionally confirmed the presence of the TiB_2_ (JCPDS 00-035-0741) phase. Philpot et al. [30] in their research have shown that the formation of nickel aluminides by the method of thermal explosion occurs sequentially, starting from the richest in aluminium and ending with the compound dominant in the product. The compounds are formed in both solid and liquid reactions, where the relative contribution of each depends on the heating rate of powders to the reaction temperature.

### 3.2. Microstructure

As a result of microscopic examinations of the sintered matrix and composite, a significant difference has been revealed in the structure of the two-phase matrix material (Figure 3, Figure 4 and Figure 5). A dendritic solidification microstructure, similar to that described by [31], was observed in the composite (Figure 3b and Figure 5). In spite of the fact that individual phases were easy to distinguish, in each case they had a different shape and character. The EDS (Energy Dispersive X-Ray Spectroscopy) analysis (Figure 4b and Figure 5b) and microhardness measurements allowed for the identification of individual phases.

The bright Ni_3_Al phase contained less aluminium than the dark NiAl phase (BSE micrographs in Figure 4a and Figure 5a), and this was also confirmed by other studies [32,33,34]. The size of the grains was similar (25–20 μm on average), but the Ni_3_Al phase distributed along the boundaries of these grains had in each case a different width—larger in the composite (about 2.5 μm) than in the sintered matrix material (about 1 μm). On the other hand, the fraction of the Ni_3_Al phase that was not included in the grain boundary region was located in the form of non-uniform precipitates inside the grains in the sintered matrix material (Figure 4c) The occurrence of this type of microstructure (defined as a Widmanstätten structure) in Ni-Al alloys was confirmed and described in [10]. The presence of such precipitates was not traced in the composite microstructure with a few exceptions in the form of an incomplete (residual) structure caused by partial transformations (incomplete synthesis) that contributed to its formation (“islands” in Figure 5c).

Therefore, it can be assumed that the factor responsible for the occurrence of different types of microstructure is the different forms of heat dissipation and hence the different mechanism of phase transformations, due to the presence of the ceramic particles of TiB_2_, even if they are added in a very small amount. These particles, evenly distributed in the mixture and then evenly deposited on the boundaries of the composite particles (Figure 3b and Figure 5b), during the sintering process enhanced by the rapidly running CS reaction, change the way in which the heat of the reaction is dissipated. Based on the results of experiments, Fan et al. [35] proposed “a dissolution-precipitation” mechanism of the combustion synthesis of NiAl and developed a model corresponding to this mechanism. It has also been shown that the incompleteness of the synthesis is associated with the use of coarse powders of the reagents.

The presence of different types of microstructure in the Ni-Al system and two-phase NiAl (β)/Ni_3_Al (γ’) alloy has already been reported in [32,36,37]. The description mainly included the lamellar, reticulated and Widmanstätten microstructure, obtained as a result of hardening and aging with alloy additions, such as iron, titanium, silicon or niobium. Kim [25] showed the microstructures of Ni-34Al and Ni-33Al-3Ti alloys after final aging, revealing the precipitates of different shapes. This phenomenon was explained by the fact that each alloy underwent a different phase transformation process during the heating up of martensite from room temperature. Wen et al. [38] have mentioned yet another type of microstructure observed when the Ni/Al molar ratio was 2:1. It was described as a relatively complex one, composed of radially running strips similar to chrysanthemum, rotating around one ridge in an ordered crystalline system.

Song et al. described the phase transformation processes from β’-martensite upon heating. The microstructure was found to be dependent on the temperature of austenite start (As), which caused the formation of a different two-phase β + γ’ microstructure. In terms of the As temperature and type of ternary constituents, the microstructure of two-phase β-NiAl/γ’-Ni_3_Al alloys was divided into three types, i.e., lamellar, reticulated and Widmanstätten. For As > 523 K, a lamellar microstructure, similar to the morphology of martensitic plates, was obtained. As< 293 K produced a typical Widmanstätten microstructure. At 293 K < As < 523 K, two different microstructures were obtained, depending on the type of the additive introduced. Reticulated microstructure was the result of the addition of Ti or Nb, which made the β → Ni_5_Al_3_ phase transformation occur more rapidly than in the case of the Si addition producing a typical Widmanstätten microstructure [37].

The type of the obtained microstructure is important because, as demonstrated by Misra, the mechanical properties of the β + γ’ alloy strongly depend on the shape of the γ’ phase formed in the β-matrix. These results additionally indicate that fine distribution of the γ’ inclusions in the β-matrix can improve the mechanical properties of a two-phase alloy [24].

### 3.3. Evaluation of the Properties of the Produced Sinters

#### 3.3.1. Density, Hardness and Young’s Modulus

The results of the measurements of density and porosity (Figure 6a) did not reveal any significant differences in these properties, which would depend on the matrix structure (relative density above 99% for both types of sinters). The results of the measurements of HV1 hardness (Figure 6b) and Young’s modulus (Figure 6c), as well as a comparison of these results with studies carried out by other authors [39,40] and with our own research [29,41], have led to the conclusion that the cause of the differences should be sought in the addition of TiB_2_ hard particles and related hardening effect rather than in the different matrix microstructures observed in these sinters.

From the comparison of the test results with [42,43,44], it follows that the use of the FAST/SPS sintering technique enables us to obtain the finished sintered product characterized by a high degree of sintering (high density and low porosity) in a shorter time and at a lower temperature. This not only saves energy, but also reduces the risk of grain growth due to a lower temperature and shorter operation time. Therefore, this method of production is particularly useful in the manufacture of composites, as it contributes to the formation of a more efficient bond at the sintered product phase boundary, especially in the case when the introduced phases differ as much in quality as the ceramic–metal systems [45,46].

In the conducted tests, the occurrence of the titanium boride phase located at the grain boundaries (Figure 7) has been identified and observed many times. The size and irregular shape of this phase indicate the occurrence of high temperature in this area during the sintering process (as a result of microspark discharges in the gap between adjacent powder particles). This possibility was pointed out many times by other researchers [47,48,49], and also in publications on reactive SPS [50,51,52].

As a result of the research, a high relative density was obtained, similar to the relative density obtained in the case of high-pressure high-temperature (HPHT) sintering [29]. The relative density was in both cases above 99%, which means that it was higher than in the SPS tests where it was 98%, but the reacted powder was used then [41]. The sintering time and temperature were 10 min and 1473 K, respectively. Compared to these tests, slight differences were noticed in the results of hardness and Young’s modulus.

#### 3.3.2. Room-Temperature Compression Tests

Contrary to common opinion (see: Introduction) on the brittleness of phases from the Ni-Al equilibrium system, as a result of uniaxial compression tests carried out at room temperature, a significant deformation of the tested samples was obtained (Figure 8 and Table 2). To some extent, it may serve as a confirmation of the statement [53] that brittleness at grain boundaries is effectively suppressed by the precipitation of the Ni_3_Al (γ’) phase along the NiAl (β) grain boundaries or in the matrix [26]. This effect occurs in the manufactured materials, as indicated by the microstructural analysis performed earlier (Section 3.2). In addition, the use of the SPS reactive sintering method can contribute to this phenomenon, similar to the case described in [54], where CS and hot pressing (CSHP) technique were used with satisfactory results to produce NiAl. The results of the compression tests are shown in Figure 8a.

Permanent (plastic) deformation Ac has reached the value of about 34% (Table 2). The value obtained for the composite was lower (approximately 25%), and it was the result of the addition of a hard and brittle TiB_2_ phase. The addition of TiB_2_ in an amount of 4 vol% caused an increase in the offset yield strength *σ*_c0.2_ from about 1150 MPa for the matrix to about 1340 MPa for the composite. The compressive strength measured as a maximum force F divided by the initial cross-sectional area was 3207 MPa and 3079 MPa, respectively. The higher compressive strength of the matrix was due to the higher deformability of the material. However, as shown in the diagram (Figure 8a), the maximum true stress was higher for the composite. The test results and the presented diagram show that after reaching the maximum true stress, the value of the stress was gradually decreasing, but no sudden destruction, as observed in other studies [7,26,55,56], took place in this particular case. This was also confirmed by observations of scrap samples after the conducted tests. A fracture in the matrix material is shown in Figure 8b. The fracture mode indicates the ductile–brittle characteristics of the material under study. With progress in plastic deformation, there was slip, followed by the formation and propagation of microcracks, ending in the brittle fracture, which occurred in the final stage of the deformation process. The fractures in composites were more brittle than in the matrix material. Observations were also carried out on the properly prepared flat side surfaces of samples with a width of about 2 mm.

In this way, it was possible to trace the deformation mechanisms operating in the materials subjected to compression tests. The samples were deformed to an appropriate level of strain. Figure 8c shows, as an example, a photo of the microstructure that has occurred in the tested composite at a strain level of about 0.35, i.e., at the maximum level of pre-failure strain. A significant degradation of the sample is observed with numerous slip bands and lines intersecting in the areas of grains and phases. This is particularly visible in the Ni_3_Al phase, but mainly because of its bright colour. The Ni_3_Al phase was also harder and more brittle (a fact confirmed by previous studies [57]), which resulted in the formation of microcracks within the area of the phase itself or at the NiAl phase boundary, leading to the propagation of cracks and, finally, to the destruction of material.

On the other hand, faster destruction of composite samples was due to the presence of the introduced TiB_2_ particles, as confirmed by the EDS observations and analyses. The destruction process of the tested material started in the places where these particles were present. As a result of tests and observations carried out at ambient temperature, it can be concluded that the material used for the matrix is characterized by high strength properties, which are not observed in the individual components of this phase when they occur alone. As can be seen in the photos taken by both scanning microscopy and optical microscopy, the Ni_3_Al phase acts, in a sense, as a reinforcement of the NiAl phase (matrix). In addition, it effectively suppresses brittleness, which translates into a high value of both compressive stress and strain. A small addition of TiB_2_ ceramic particles slightly increases the brittleness of the produced composite (lower plastic deformation) with a simultaneous increase in the value of the offset yield strength.

As a result of the compression tests carried out at room temperature, it was shown that the matrix material was characterized by a high compressive strength (3207 MPa) and high plastic deformation (34.2%). In the studies carried out by Ma et al. [58], the compressive strength for single phase NiAl alloy and two-phase NiAl alloys was 535 MPa and 2124 MPa, respectively. The yield strength and plastic deformation were also higher for the material we tested, produced by reactive SPS. The research by Ma et al. also achieved the effect of improving plastic properties by creating a two-phase NiAl-Cr(Mo) microstructure. This was only a confirmation of our results, where the production of a two-phase material gives a synergistic effect in the form of higher strength properties and plastic properties compared to the single phase NiAl and Ni_3_Al.

#### 3.3.3. High-Temperature Compression Tests

As a result of the high-temperature compression tests (Figure 9a), regardless of the temperature, a nearly identical course of the deformation curves was observed for the selected group of materials (matrix–composite). For the matrix material (two-phase alloy), at a given test temperature, after reaching the offset yield strength σ_c0.2_, a much faster systematic increase in the value of the plastic flow stress was observed. With only a slight difference in the strengths of the matrix and the composite, this caused the graphs to intersect. Additionally, as the deformation temperature was increasing, the graphs were observed to intersect in a much shorter time, i.e., at a lower strain. The change in the occurrence of the maximum true stress was observed to follow the same principle. The higher the test temperature, the shorter the time to reach this value, i.e., the lower the strain. Hence, at lower temperatures, the deformation range, which resulted in the material hardening, was smaller compared to high temperatures. Only the base alloy deviated from this rule, being deformed at a temperature of 1073 K and with the value obtained constantly increasing (hardening was observed to continue up to the strain value of 1.12—Figure 9a, upper part). This indicates the high plasticity of this alloy, as has also been demonstrated by other studies [9], where the plasticity above 100% was achieved in the tensile tests at a slightly higher temperature (1273 K).

At each test temperature, the composite material was characterized by the offset yield strength σ_c0.2_ higher than the dual-phase matrix material (Table 2). Depending on the temperature, the decrease in this value was at first insignificant (to 1270 MPa for the composite and to about 1130 MPa for the matrix at the test temperature of 873 K). With the increasing temperature (1073 K), this decrease became more prominent, reaching values of 480 MPa and 390 MPa, respectively. This means that the distinct “softening” point (brittle-to-ductile transition temperature—BDTT) lies somewhere above 873 K, contrary to the results of the studies carried out by Zeumer et al. [59], where the temperature at a similar compressive strain rate was about 100 K lower. Similar results were also obtained by R.S.Chen et al. [60].

The samples deformed at room temperature and at 823 K failed in a shorter time, and hence the compressive strength (σ_c_) was determined for them. The course of the deformation curves at higher temperatures shows that after exceeding the maximum stress, the value of the flow stress decreases, first, and then gets stabilized at a certain level. It means that the dynamic processes of structure rebuilding (dynamic recovery and dynamic recrystallization) take place in the material, as demonstrated in [55,61,62,63].

Due to the dependencies presented above and to show changes in the properties of the examined materials more clearly, the compressive strength presented in Table 2 as σ_cmax_ was determined for the maximum values of true stress reached at the specific levels of strain (indicated in the table).

Microstructural studies of samples deformed at elevated temperatures showed numerous intersecting slip lines and bands in the grain area of both the matrix material (Figure 9c) and composite (Figure 10b). Fractures and cracks (Figure 9b and Figure 10a) in the samples deformed at 873 K are still of a transcrystalline cleavage mode. This type of fracture is the result of cracks propagating along certain crystallographic planes of the grain. The photos show that the surface of the fracture or crack has characteristic discontinuities in the form of faults, forming a system of intersecting “river basins” (fault zone). The occurrence of discontinuities of this type means that the development of cracks did not occur along one crystal plane, but a step change in the plane of their propagation took place. Figure 9d and Figure 10a are an additional confirmation of the obtained high density, i.e., about 99% of the theoretical density (Figure 6a). Thus, they indicate that the choice of reactive sintering as a method for the manufacture of this type of material was correct. Cracks in the tested materials run through the grains (transcrystalline mode of cracking), which indicates the high strength of the grain boundaries. The samples did not fail at high temperatures, i.e., above 873 K.

Studies of the composite subjected to deformation have additionally revealed that the deformation process was blocked by the slip and piling up of individual moving microstructural elements on the ceramic particles introduced into the matrix (Figure 10c). Another effect was oxidation of the above elements (layers).

As a result of the compression tests carried out at elevated temperatures, high strength properties (offset yield strength) and high plastic properties were observed. The yield strength results obtained by Chen et al. [60] (multi-phase NiAl alloys) were, on average, by about 100 MPa lower at each test temperature.

#### 3.3.4. Tribological Properties

The ball-on-disc abrasion tests have demonstrated that the addition of 4 vol % TiB_2_ to the two-phase NiAl/Ni_3_Al matrix results in a significant improvement of abrasion resistance, as evidenced by the wear coefficient (Figure 11a), reduced depth (Figure 11b) and width of the wear track, and lower relative mass loss (25% improvement).

The NiAl/TiB_2_ composite is characterized by a lower coefficient of friction than its respective matrix, i.e., 0.48 against 0.55 (Figure 11a). From the wear rate graph (Figure 11b), it follows that in both materials the wear rate is the highest in the first few minutes of the test, and the wear is more severe and changes in a more inconsistent way in the sintered matrix. This is most probably the effect of lapping and/or oxide formation. Later, in both cases, the wear tends to stabilize.

Making a comparison with our previous research [29], where the dual-phase matrix was produced before the sintering process, with the same test parameters, similar results of the wear behaviour were obtained. A slight difference in the results of the friction coefficient and the wear coefficient (specific wear rate) may result from the different process of the manufacture of the dual-phase matrix and/or from the use of a different HPHT sintering method (7.5 GPa, 1779 K). The obtained value of the wear coefficient is still much lower than the value obtained for 316L austenitic stainless steel and composite made by SPS [64], C45 steel (AISI 1045) [65], NiAl alloy or NiAl self-lubricating matrix composite with the addition of graphene (improving the tribological properties of the composite) made by SPS [66].

Studies of the wear track (Figure 12a) and EDS analysis (Figure 12b) indicate that, compared to the starting condition where oxides were not found, after 10,000 s of the ball-on-disc test, the amount of oxides on the surface is already quite significant. As described in numerous articles [31,67,68,69], due to their specific properties, oxides affect the behaviour of the material and cause differences in its abrasive wear under the conditions of dry friction. The formation of slippery oxides in the friction process can explain the observed reduced wear.

Studies of Al_2_O_3_ balls (Figure 13) and measurements of their mass after friction tests have not shown any loss of material, so it is unlikely that it could remain on the surface of the produced sinters, giving rise to the presence of oxides, though such a possibility cannot be totally ruled out. On the surface of balls, seizures formed as a result of the sintered material transfer and slight increase in the mass of these balls (within the limits of statistical error) were noticed.

Characteristic surface changes, such as torn out TiB_2_ particles resulting from the ball-on-disc friction and causing surface degradation through fissuring and wear (Figure 12a) or seizing (Figure 14b), were additionally observed. The even wear track visible on the surface of sinters additionally indicates the high resistance to wear of the intermetallic material (Figure 14a), and particularly of the composite based on its matrix.

The conducted research has confirmed the advantages of the application of the reactive SPS method in the fabrication of two-phase Ni-Al materials and composites based on their matrix reinforced with TiB_2_. Compared to other sintering methods, the use of this method gives the material a compact microstructure, free from pores, characterized by higher strength and better tribological properties. Compared to traditional SPS sintering, the duration of the process is shorter (reduced from 10 min to 30 s), and in the case of the above-mentioned HPHT sintering, by using a lower temperature and pressure (1273 K and 48 MPa, respectively), we save the energy needed to produce the material.

## 4. Conclusions

The results of the studies of the physical, mechanical and tribological properties show that the addition of 4 vol% TiB_2_ to the composite matrix improves these properties, and despite significant differences, the obtained values of these parameters depend on the particle hardening effect rather than on the difference in matrix microstructure observed during tests. Even at room temperature, both the dual-phase matrix material and composite based on this material were characterized by high plastic deformation (about 34% and 28%, respectively) and high compressive strength—above 3 GPa. In the temperature range of 973–1073 K, a high level of deformation was observed (for the matrix material even a transition to the state of high ductility was observed) with still a satisfactory compressive strength (over 500 MPa at the highest test temperature). The determined brittle-to-ductile transition temperature (BDTT) was above 873 K for both tested materials. Due to the simultaneous application of FAST/SPS and CS, it is possible to carry out the process at a lower temperature and in a shorter time, which has a significant impact on the high quality of the obtained sinters with elimination of the grain growth phenomena.

## Figures and Tables

**Figure 1 materials-13-05668-f001:**
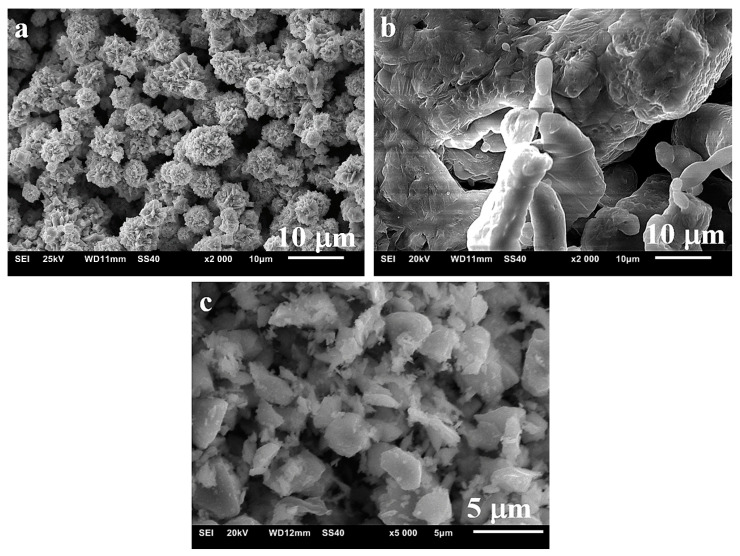
Powder morphology: (**a**) nickel, (**b**) aluminium, (**c**) TiB_2._

**Figure 2 materials-13-05668-f002:**
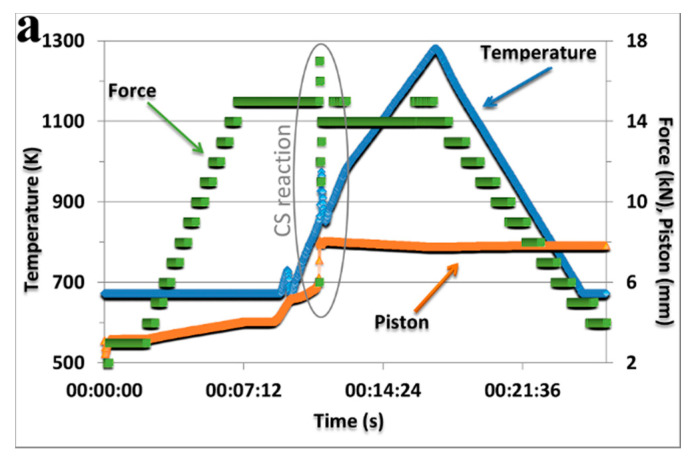
(**a**) Diagram of the course of the NiAl alloy (matrix material) sintering process, (**b**) the results of XRD analysis of both matrix material and composite.

**Figure 3 materials-13-05668-f003:**
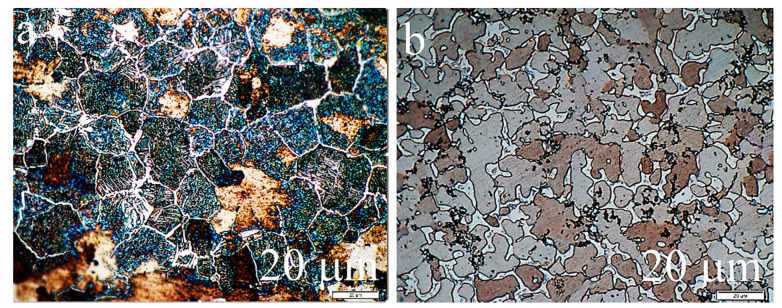
Microstructure of: (**a**) matrix material and (**b**) composite—etched surface, Nomarski contrast, optical microscope.

**Figure 4 materials-13-05668-f004:**
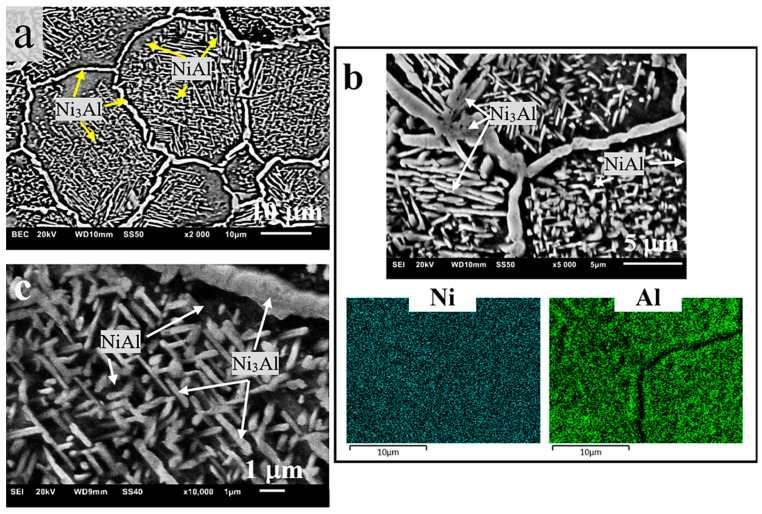
Microstructure of the NiAl sinter at various magnifications: (**a**) BSE (backscattered electrons) micrographs, (**b**) EDS mapping analysis, (**c**) non-uniform intragranular precipitates of Ni_3_Al.

**Figure 5 materials-13-05668-f005:**
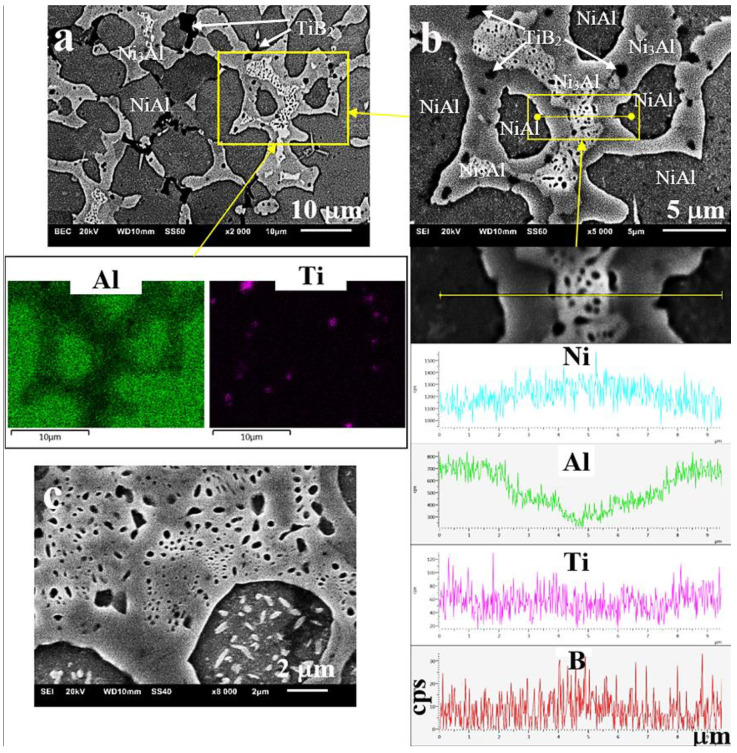
Microstructure of the NiAl/TiB_2_ composite at various magnifications: (**a**) BSE micrographs, (**b**) EDS analysis of the distribution of individual microstructural constituents, (**c**) “islands” of the residual Widmanstätten structure.

**Figure 6 materials-13-05668-f006:**
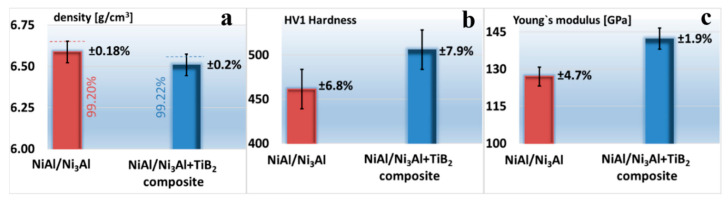
Comparison of selected properties of the sintered matrix material and composite: (**a**) density, (**b**) hardness, (**c**) Young’s modulus.

**Figure 7 materials-13-05668-f007:**
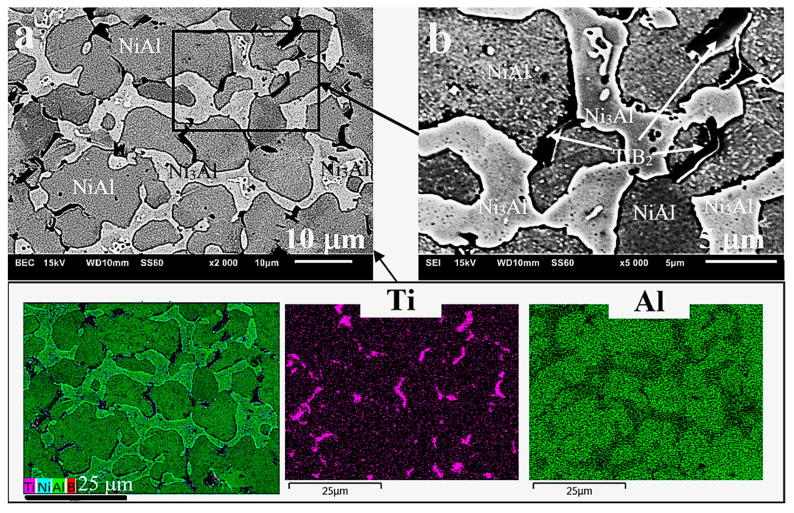
Microstructure of the NiAl/Ni_3_Al + TiB_2_ composite; analysis of the (**a**) distribution (EDS map identification) and (**b**) shape of TiB_2_ particles.

**Figure 8 materials-13-05668-f008:**
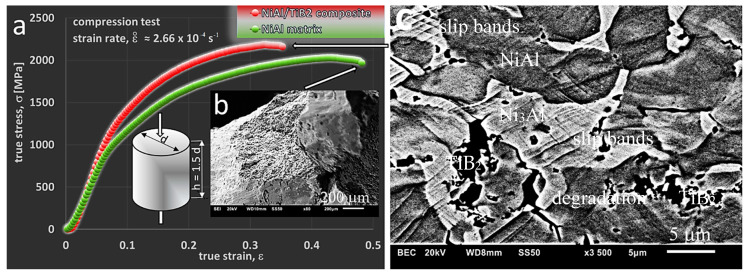
(**a**) Compressive stress–strain curves of the dual-phase NiAl/Ni_3_Al matrix and the NiAl/Ni_3_Al + TiB_2_ composite, specimens with dimensions h × d (scheme) were deformed at room temperature at the initial strain rate, (**b**) brittle–ductile fracture in the NiAl matrix sample (after destruction), and (**c**) microstructure observed on the flat side surface of the NiAl + TiB_2_ composite sample (before destruction) after the compression test; the direction of compression as indicated in the sample scheme.

**Figure 9 materials-13-05668-f009:**
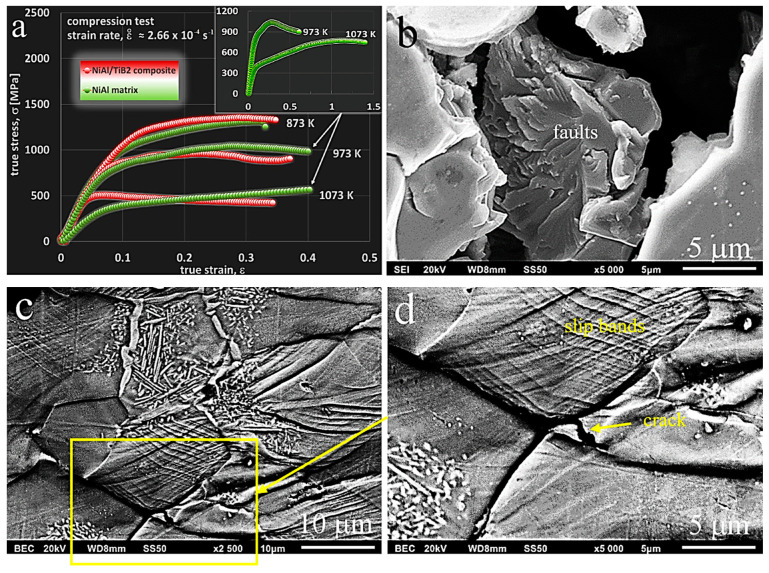
High-temperature compression test: (**a**) course of curves at specific temperatures of NiAl matrix and composite with additions of TiB_2_; matrix microstructure (**b**) after deformation at a temperature of 873 K, (**c**) at a strain level of 0.3, (**d**) cracks at grain boundary.

**Figure 10 materials-13-05668-f010:**
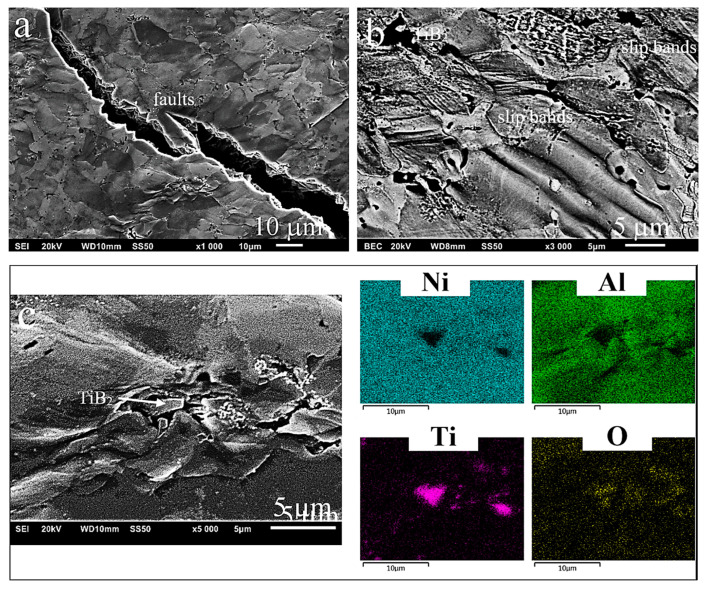
High-temperature compression test: composite microstructure (**a**) after deformation at a temperature of 873 K, cracks in the form of faults, (**b**) at a strain level of 0.3, (**c**) blocking of slip and piling up of grains on the particle of TiB_2_, map identification of individual microstructural constituents, including TiB_2_ particles and oxides.

**Figure 11 materials-13-05668-f011:**
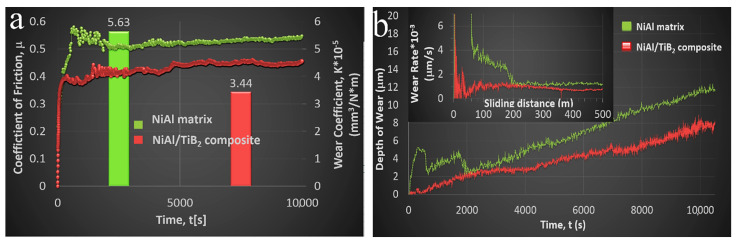
The course of the relationship between: (**a**) coefficient of friction and wear coefficient, (**b**) depth of scar and rate of wear as a function of the ball-on-disc process duration and sliding distance, respectively, observed in the manufactured sinters.

**Figure 12 materials-13-05668-f012:**
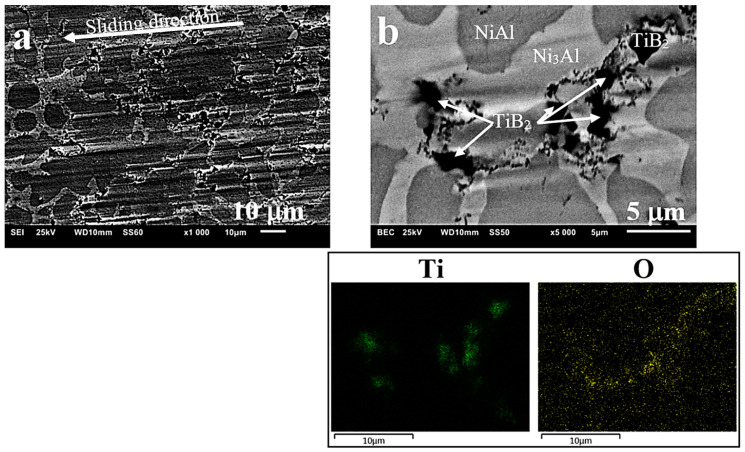
Composite surface after the ball-on-disc friction process; (**a**) surface degradation with TiB_2_ particles removed, (**b**) embedded TiB_2_ particles and oxidized products (oxides formed)—EDS analysis.

**Figure 13 materials-13-05668-f013:**
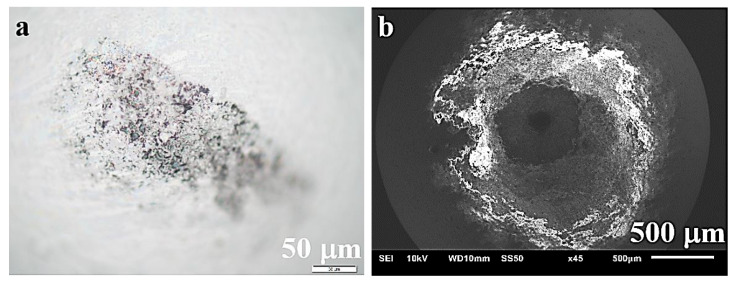
Ball surface at the point of friction/contact with the composite surface after the ball-on-disc friction process: (**a**) optical microscope, (**b**) SEM.

**Figure 14 materials-13-05668-f014:**
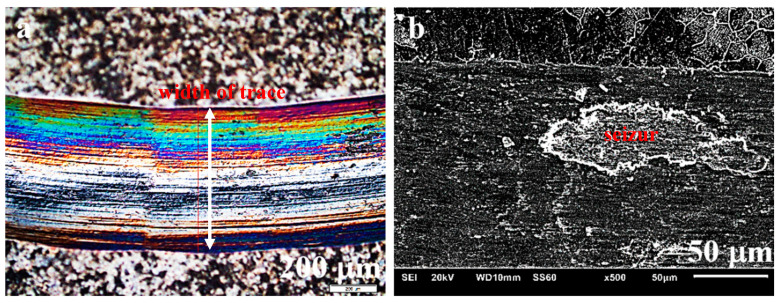
Surface of the NiAl sinter showing: (**a**) track (path) of ball travel in the ball-on-disc friction process, and (**b**) seizure resulting from this process.

**Table 1 materials-13-05668-t001:** The ball-on-disc test parameters.

Load Applied [N]	Radius of the Wear Track [mm]	RPM	Ball Parameters	Chamber Parameters			Test Duration [s]
				Temp. [K]	Humidity [%]	Medium	
5.0	5.0	192.0	Material—Al_2_O_3_ Diameter-1/8” (in)	298	38.2	Air	10,000

**Table 2 materials-13-05668-t002:** Strength properties of the tested materials obtained in a compression test carried out at room temperature and at elevated temperatures.

Material	Test Temperature [K]	Compressive Offset Yield Strength σ_c0.2_ [MPa]	Compressive Strength for Max. True Stress/def. Level σ_cmax_ [MPa]/ɛ_c_	Compressive Strength σ_c_ [MPa]	Plastic Deformation A_c_ [%]
**NiAl/Ni_3_Al matrix**	296	1150	3105/0.43	3207	34.2
	873	1130	1830/0.32	1842	25
	973	800	1395/0.28	not destroyed	not destroyed
	1073	390	2377/1.12	not destroyed	not destroyed
**NiAl/Ni_3_Al + TiB_2_ composite**	296	1340	3024/0.33	3079	27.8
	873	1270	1816/0.29	1891	23.9
	973	890	1210/0.22	not destroyed	not destroyed
	1073	480	534/0.07	not destroyed	not destroyed

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
