# Peer review of "Effect of Reactive SPS on the Microstructure and Properties of a Dual-Phase Ni-Al Intermetallic Compound and Ni-Al-TiB2 Composite"

_materials, 2020, doi:10.3390/ma13245668_

Round 1

Reviewer 1 Report

Please find my comments attached.

Reviewer 2 Report

This is regarding the manuscript titled : The effect of using reactive SPS on microstructure and properties of a dual-phase Ni-Al composite matrix

The paper describes and compares the microstructure, mechanical and tribological properties of two Ni- Al alloys prepared by SPS. There is a detailed description of experiment along with results and thoughtful discussion. The paper is of high quality and worth of publication. However, there is some missing information or inaccuracy in the text:

- please change the abbreviation BDTT in abstract to the full text ductile-to-brittle transition temperature
- could you add the RPM value of milling to the text along with the used ball to powder ration and used atmosphere?
- line 62 and 63, " ...CS reaction taking place in the sintered ..." it sounds that CS was performed after, not during SPS, i recommend to change this part of the sentence to :" ...CS reaction taking place during SPS in the nickel-aluminium alloy..."
- line 106 - scanning electron microscope
- Figure 2b- could you change the Y axis to log scale so as the peaks are more visible?
- Where are those EDX maps in Figure 5 from? Are these from a given region of Fig 5a? If so please denote that region, or include the corresponding SE/BSE figure
- line 137: "size of the grains" (and on other places)- not grains, but former powder particles, or do you think there is just one grain inside a former powder particle?
- Caption and reference in the text is missing to the lower part of Figure 7
- line 213-216:" characteristic shape of the TiB2 phase " what is the characteristic shape of this phase? how it indicates the high temperature and how high should be that temperature be?
-line 280-281: "Additionally, as the deformation temperature was increasing, the graphs were observed to intersect in a much shorter time, i.e. at a lower strain."- but Figure 9a shows the intersection of graphs with increasing temperature at increasing strain, or am I missing something?
-Figure 10 - from what region of Figure A,b or c are those EDX maps from?

Round 2

Reviewer 1 Report

There have been corrections applied by the authors, however again some mistakes need rectification before acceptance of the manuscript.

  1. Figure 4 contains paragraph symbol in labelling of the phases. This should be removed. The contrast of text in the images is sometimes vague, please correct that.
  2. Again the problem with different fonts of different size has not been resolved. This is not an issue with the publisher but the authors should make the effort of legibly constructing the manuscript. As an example, line 121 - 126, line 153 - 155, line 165 - 169, line 362 - 364, line 377 - 379 etc. This is not good practice in a proofread version.
  3. Figure 2 (b): The legend in XRD plots is not legible with text merging in the figure. What is the phase present in Ni-Al matrix at major peak 66 - 67 degrees, Ni-Al base? Explain the reason behind reduction in the major peak intensities in the composite with the addition of TiB2 and the composite peaks should be assigned on the top rather at the bottom.
  4. The XRD pattern references should be included in the text.
  5. Cite these kind of sentences in the introduction:" Our aim was to use in an unconventional way an advanced method of sintering to produce a composite material based on intermetallic Ni-Al phases, being aware of the fact that little knowledge on this subject is available in the literature.", merging together the significance of SPS towards the study.
  6. Since I am the reviewer of this article, it must be presented in a systematic way which is easier applied science to follow rather than trivial to quote. Please work on this segment in the Introduction: " compare how previously addition of TiB2 has offered improvements in these intermetallic alloys and how different phase transformation contribute to synergistic effect." This is important for definition of study objectives. Even if you need to quote yourself, I would not mind if you can justify it appropriately. It is however not suitable that you say this work was done previously so we cannot justify science here because you must justify in every scientific writing in a distinct manner. So even if you are to quote yourself, justify how this work is different from the previous studies in the introduction segment.
  7. Where the goals are justified in results section, final segment of introduction part should be constructed to highlight these findings irrespective of fear that manuscript will grow longer. 
  8. Why EDS is unable to provide identification of complex phases? Explain with setup of SEM - EDS unit and the kind of phases it can precisely distinguish to a certain resolution.
  9. I would suggest the authors to proofread what they write back to the reviewer. What is the explanation to this answer?
    "As indicated in the title of the article, the main aim of the research and of the results obtained is to investigate the effect of the sintering method (reactive SPS) on the microstructure and properties of the matrix. Here, the term “matrix” can mean either the NiAl phase (in which the hard Ni3Al phase is embedded) or the NiAl/Ni3Al system which can act as a two-phase matrix for the ceramic particles. Therefore, at this stage of the studies and in this particular article, it was not the authors’ intention to take up another, more complex topic related to the effect of the introduced TiB2 particles."
    Your starting and terminal statements are contradicting since TiB2 has been used in your study, so authors should know the impact and plausible effects of TiB2 because this is science reporting and you are not making a milkshake!!!

  10. When I asked you to write about solution-reprecipitation then you were required to explain even reference from literature which phase is thermodynamically going dissolution and how it reprecipitates back in the matrix to what kind of morphology and if precipitation happens on the interior of grain and/or grain boundaries as well for excessive solute. It goes for both base material and composite with TiB2. Also mention if TiB2 has miscibility in Ni-Al system, for the same reason I asked for the formation of Ni-Ti or other phases with acicular type morphologies.
  11. Again, the authors are requested not to teach the reviewer if they find a mistake in the manuscript because of lack of proper proofreading. Secondly the answers to reviewer must answer previous questions, rather than neglecting them altogether. Manuscript must be rectified in the spirit of science so that it is legible for everyone and all the readers do not think from the same sphere where authors are entangled. Due to such a thought process, these trivial mistakes still exist in this manuscript. A student lab report may not acceptable be as a manuscript, I must suggest authors again.
  12. There are placement mistakes as mentioned above. The presentation of results in any publication go in a flow, e.g. you take one segment and then explain that part by first mentioning your observations and comparing to literature. Then finally discuss at the end or along with the results what impact these findings have compared to previous literature. I agree that the link must exist with the microstructure, but at least self-assess if you are making mistakes or not by proofreading. But in your case, few sections are not clear; for example: what is the difference between the micrographs in figure 5 and 7 are not described in the text or linked with each other. The difference may be two-phase alloy system, but the authors did not cite this difference in micrographs in the text either. Where you are discussing Figure 5, you preclude Figure 7 details as well. That's precisely how you are not supposed to write a science publication and therefore the authors were requested to review it and proofread thoroughly.
  13. Now i like how tribological part is rephrased and concluded with results properly defined. Precisely, that is how I anticipate authors had done with previous sections and proofread them before submission.
  14. Minor revisions yet thorough proofreading is recommended.
